# All In: Bridging Input Feature Spaces Towards Graph Foundation Models

**Moshe Eliasof**
University of Cambridge
Cambridge, UK
me532@cam.ac.uk

**Krishna Sri Ipsit Mantri**
Purdue University
Indiana, USA
mantrik@purdue.edu

**Beatrice Bevilacqua**
Purdue University
Indiana, USA
bbevilac@purdue.edu

**Bruno Ribeiro**
Purdue University
Indiana, USA
ribeirob@purdue.edu

**Carola-Bibiane Schönlieb**
University of Cambridge
Cambridge, UK
cbs31@cam.ac.uk

## Abstract

Graph learning is hindered by the lack of a shared input space, as features vary in semantics and dimensionality across datasets, preventing models from generalizing. We propose ALL-IN, a method that enables transferability across these diverse input feature spaces. Our approach projects node features into a shared random space and builds representations from covariance-based statistics, removing dependence on the original feature space. Theoretically, we show that the resulting node representations are invariant in distribution to input feature permutations and the expected operator is invariant to orthogonal transformations of the input features. Empirically, ALL-IN achieves strong performance on unseen datasets with new features across various tasks without requiring retraining, pointing to a promising direction for truly input-agnostic and transferable graph models.

## 1 Introduction

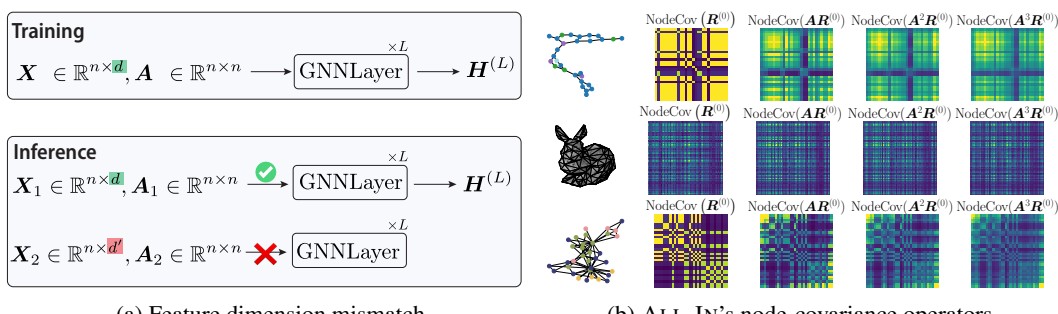

(a) Feature dimension mismatch      (b) ALL-IN's node-covariance operators

Figure 1: **ALL-IN enables feature-agnostic learning.** (a) Standard GNNs fail when feature dimensions differ. (b) ALL-IN computes node-covariance matrices from stochastic projections, yielding a consistent, feature-agnostic operator.

Foundation models in vision and language benefit from shared input spaces, enabling effective transfer. In contrast, graph learning faces a major barrier: graphs from different domains often have incompatible node feature spaces differing in dimension, semantics, and distribution making it

Preprint.

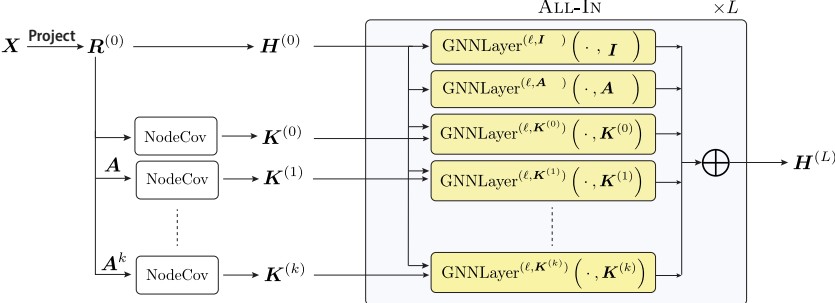

Figure 2: **ALL-IN Architecture.** Input features are randomly projected and used to compute propagated node-covariance operators $\{K^{(p)}\}_{p=0}^{k}$ via propagation. These operators drive GNN layers, whose outputs are concatenated for final representations.

difficult to build models that generalize across datasets. While recent efforts leverage LLMs via graph serialization [31, 53] or align features through projections [46, 52], these methods either discard structural detail or require careful adaptation, limiting scalability and generality.

We propose ALL-IN (All Input spaces), a statistical approach that sidesteps feature heterogeneity by modeling node features as samples from a latent distribution. Instead of operating on raw features, ALL-IN computes a stochastic node-covariance matrix from random projections, capturing pairwise node similarities in a way that is invariant to feature permutations, orthogonal transformations, and dimensional mismatches (Figure 1). This covariance-based operator serves as a robust, input-agnostic foundation for GNNs, enabling transfer across graphs with entirely different feature spaces. We demonstrate theoretically and empirically that ALL-IN preserves task-relevant structure while supporting strong cross-dataset generalization, offering a promising path toward true graph foundation models.

## 2 Method

Our method, ALL-IN, enables transfer across graphs with heterogeneous input features by replacing raw node features with *covariance-based operators* that capture robust, general-purpose node relationships. The framework consists of three stages: (i) Random Feature Projection into a shared space, (ii) Node-Covariance Operator computation to model feature-driven node similarities, and (iii) Operator-based GNN propagation for representation learning (Figure 2).

Given node features $X \in \mathbb{R}^{n \times d}$, we first we first project them via a random isotropic matrix $C \in \mathbb{R}^{d \times h}$:

$$R^{(0)} = XC, \qquad \text{with } \text{vec}(C) \sim \mathcal{N}(0, I_{dh}), \tag{1}$$

sampled independently at each forward pass. This ensures distributional invariance to arbitrary input feature permutations, a critical property for generalization across datasets with differing feature orderings. We then compute *node-covariance operators* to capture second-order node similarities. After centering $R_c^{(0)} = R^{(0)} - \mathbf{1}_n \bar{r}$ with $\bar{r} = \frac{1}{n} \sum_i^n R_i^{(0)} \in \mathbb{R}^{1 \times h}$, we define:

$$K^{(0)} = \text{NodeCov}(R^{(0)}) = \frac{1}{h} R_c^{(0)} R_c^{(0)T} \in \mathbb{R}^{n \times n}. \tag{2}$$

To incorporate structural context, we propagate features via $R^{(p)} = A^{(p)} R^{(0)}$ and compute higher-order operators $K^{(p)} = \text{NodeCov}(R^{(p)})$ for $p = 1, \cdots, k$. These operators encode feature co-variation across increasing graph neighborhoods. The final set of propagation operators includes identity, adjacency, and covariance matrices:

$$\mathcal{O} = \{I, A, K^{(0)}, K^{(1)}, \ldots, K^{(k)}\}. \tag{3}$$

Initial node representations are $H^{(0)} = R^{(0)} \oplus S$, where $S \in \mathbb{R}^{n \times h_s}$ is a structural encoding matrix. At layer $\ell$, we update representations via:

$$H^{(\ell)} = \bigoplus_{O \in \mathcal{O}} \text{GNNLayer}^{(\ell, O)}(H^{(\ell-1)}, O), \tag{4}$$

Table 1: Performance of ALL-IN on pre-training source datasets compared to specialized supervised baselines trained individually per dataset. ALL-IN maintains highly competitive performance.

| Method | ZINC (MAE ↓) | MOLHIV (ROC-AUC ↑) | MOLESOL (RMSE ↓) | MOLTOX21 (ROC-AUC ↑) | MNIST (ACC ↑) | CIFAR10 (ACC ↑) | MODELNET (ACC ↑) | CUNEIFORM (ACC ↑) | MSRC 21 (ACC ↑) |
|---|---|---|---|---|---|---|---|---|---|
| GCN [26] | 0.3674 | 76.06 | 1.11 | 75.29 | 90.120 | 54.142 | 17.18 | 45.67 | 89.53 |
| GAT [43] | 0.3842 | 76.00 | 1.05 | 75.21 | 95.535 | 64.223 | 65.20 | 78.60 | 82.10 |
| GIN [47] | 0.1630 | 75.58 | 1.17 | 74.91 | 96.485 | 55.255 | 73.13 | 79.05 | 86.31 |
| ALL-IN (0 props) | 0.1557 | 72.74 | 1.28 | 68.19 | 94.57 | 40.11 | 37.11 | 89.88 | 97.51 |
| ALL-IN | 0.1237 | 74.49 | 1.29 | 68.20 | 95.22 | 40.08 | 39.37 | 91.17 | 98.08 |

using learnable transformations per operator. This design allows ALL-IN to operate in a feature-agnostic manner, enabling transfer to graphs with entirely new input spaces.

For edge features, an analogous projection and aggregation strategy yields edge-derived covariance operators, which are added to $\mathcal{O}$ preserving compatibility across edge feature spaces.

**Transferability via Expected Operator Invariance.** The theoretical foundation for cross-dataset generalization lies in the invariance and consistency of the expected node-covariance operator. The following theorem shows that, on average, ALL-IN recovers a basis-invariant representation of node similarities enabling stable transfer across datasets with different feature parameterizations.

**Theorem 2.1** (Expected Invariance to Orthogonal Transformations). *Let $X \in \mathbb{R}^{n \times d}$ be node features, $Q \in \mathbb{R}^{d \times d}$ be an orthogonal matrix, and $h$ be the projection dimension. Consider a random projection matrix $C \in \mathbb{R}^{d \times h}$ with $\mathrm{vec}(C) \sim \mathcal{N}(0, I_{dh})$. Let $NodeCov(R^{(0)}) = \frac{1}{h}(\Pi_c R^{(0)})(\Pi_c R^{(0)})^T$ be the Node Covariance operator (Equation (2)), where $\Pi_c = I_n - \frac{1}{n}\mathbf{1}_n\mathbf{1}_n^T$ is the centering matrix. Then, the expected Node Covariance computed from the stochastically projected features is invariant to the orthogonal transformation $Q$:*

$$\mathbb{E}_C[NodeCov(XQC)] = \mathbb{E}_C[NodeCov(XC)] = \Pi_c X X^T \Pi_c \tag{5}$$

*where the expectation $\mathbb{E}_C[\cdot]$ is over the random sampling of $C$, and $\Pi_c X X^T \Pi_c$ is the Gram matrix of the centered original features.*

This result ensures that ALL-IN learns representations based on intrinsic data geometry rather than arbitrary feature representations. When graphs share underlying structural patterns (e.g., homophily, role structure), their expected operators align, even with different features, enabling effective transfer. Moreover, as $h \to \infty$, the stochastic operator converges to its expectation (Proposition B.5), ensuring reliable estimation in practice.

## 3 Experiments

We evaluate ALL-IN 's ability to learn transferable graph representations and, critically, to generalize to unseen datasets with entirely new input features. Our experiments address:

(**Q1**) How does a single ALL-IN model perform on diverse source datasets compared to specialized GNNs trained per dataset?

(**Q2**) Can ALL-IN transfer effectively to new datasets with novel features and tasks?

We present main results here; additional experiments (including runtime) and implementation details (datasets, hyperparameters) are in Appendices D to F.

### 3.1 Performance on Pre-training Source Datasets (A1)

We pre-train a single ALL-IN encoder on nine diverse graph datasets, spanning molecules(ZINC [10], OGBG-MOLHIV [22], OGBG-MOLESOL [22], OGBG-MOLTOX21 [22]), images(MNIST [10], CIFAR10 [10], CUNEIFORM [32], MSRC 21 [32]), and 3D shapes (MODELNET [45]) with heterogeneous features and tasks. For each dataset-task pair, a dedicated prediction head is attached to the shared ALL-IN encoder. We compare against GNN baselines (GCN [26], GAT [43], GIN [47]) trained individually for each dataset, using their original, dataset-specific input features. These supervised baselines are thus specialized for each respective dataset.

**Results.** Specialized GNNs are expected to perform well on individual source datasets due to task-specific training. For ALL-IN, the goal is to match them using a single shared encoder demonstrating general-purpose representation learning without significant performance loss.

As shown in Table 1, ALL-IN is broadly competitive: it outperforms all baselines on ZINC (MAE 0.1237 vs. GIN 0.1630), and achieves substantial gains on textscCuneiform (91.17% vs.GIN 79.05%) and MSRC 21 (98.08% vs.GCN 89.53%). Only on CIFAR10 and MODELNET do dedicated models prevail. The full ALL-IN consistently surpasses ALL-IN (0 props), confirming that propagated covariance operators enhance representation learning.

These results show that a single pre-trained encoder can perform strongly across diverse dataset enabling effective multi-task pre-training.

Table 2: Node classification on unseen datasets with new features. ALL-IN performs competitively with SOTA.

| Method | CORA (ACC ↑) | CITESEER (ACC ↑) | PUBMED (ACC ↑) |
|---|---|---|---|
| **SUPERVISED BASELINES** | | | |
| MLP | 48.42 ± 0.63 | 48.56 ± 0.27 | 66.26 ± 1.53 |
| GCN [26] | 78.86 ± 1.48 | 64.52 ± 0.89 | 74.49 ± 0.99 |
| GIN [47] | 67.10 ± 3.00 | 58.80 ± 2.20 | 68.40 ± 2.70 |
| **LLM-AUGMENTED GNNs** | | | |
| OFA [31] | 76.10 ± 4.11 | 73.04 ± 2.88 | 75.61 ± 5.06 |
| GLEM-LM [8] | 67.55 ± 3.53 | 66.00 ± 5.66 | 62.12 ± 0.07 |
| **LLM-BASED** | | | |
| GRAPHTEXT [53] | 75.41 ± 2.08 | 58.24 ± 0.26 | 63.70 ± 0.29 |
| RWNN-LLAMA3-8B [25] | 72.29 | N/A | N/A |
| **GNN-BASED** | | | |
| ANYGRAPH [46] | 62.60 ± 0.14 | 19.32 ± 0.37 | 70.73 ± 4.13 |
| GRAPHANY [54] | 79.36 ± 0.23 | 68.42 ± 0.39 | 76.30 ± 0.41 |
| MDGPT [49] | 43.36 ± 8.92 | 42.50 ± 9.78 | 51.91 ± 9.00 |
| GCOPE [52] | 35.54 ± 2.09 | 31.18 ± 4.35 | 32.87 ± 4.08 |
| GPPT [38] | 43.15 ± 9.44 | 37.26 ± 6.17 | 48.31 ± 17.72 |
| ALL-IN-ONE [39] | 52.39 ± 10.17 | 40.41 ± 2.80 | 45.17 ± 6.45 |
| GPROMPT [19] | 56.66 ± 11.22 | 53.21 ± 10.94 | 39.74 ± 15.35 |
| GPF [11] | 38.57 ± 5.41 | 31.16 ± 8.05 | 49.99 ± 8.86 |
| GPF-PLUS [11] | 55.77 ± 10.30 | 59.67 ± 11.87 | 46.64 ± 18.97 |
| ULTRA (3G) [16] | 79.40 ± 0.00 | 67.40 ± 0.00 | 77.90 ± 0.00 |
| SCORE [44] | 81.80 ± 1.02 | 71.33 ± 0.27 | 82.93 ± 0.55 |
| ALL-IN (0 props) | 79.26 ± 1.08 | 65.96 ± 1.25 | 77.30 ± 0.47 |
| ALL-IN | 82.13 ± 0.97 | 69.12 ± 0.89 | 78.03 ± 0.82 |

## 3.2 Transferability
### to Unseen Datasets and Input Features (A2)

We evaluate the core claim of ALL-IN: that a single pre-trained encoder can generalize to **unseen datasets with entirely new input features**. To isolate representation quality, we keep the encoder frozen and train only a lightweight prediction head on top for each new dataset covering both node and graph tasks with novel features and labels. We compare against supervised GNNs, LLM-augmented models, and recent graph foundation models (see Appendix D for categorization). Full results on graph classification (e.g., MUTAG, PROTEINS) are in Appendix D, showing similarly strong transfer; here we focus on node classification.

**Results.** ALL-IN achieves strong performance on unseen node classification datasets (Table 2), despite never seeing their features during pre-training. On CORA, it reaches an accuracy of 82.13% which not only surpasses standard supervised GCN (78.86%), but it also exceeds leading baselines like SCORE [44] (81.80%) and GRAPHANY [54] (79.36%). Notably, ALL-IN (0 props) already performs competitively, but the full model consistently improves upon it confirming that propagated covariance operators enhance generalization.

These results demonstrate that ALL-IN learns **general-purpose representations** that transfer effectively to new datasets with novel features, supporting both node and graph tasks without task-specific design unlike specialized models such as (GRAPHANY [54], GRAPHTEXT [53], GCOPE [52], ANYGRAPH [46]), only supporting node classification.

## 4 Conclusion

Input feature heterogeneity critically limits the development of Graph Foundation Models (GFMs). Our ALL-IN offers a theoretically-grounded solution, processing arbitrary node features through stochastic projections and node-covariance operators to build robust representations independent of the original feature space. We prove that these representations achieve distributional invariance to input feature permutations, and their underlying expected operator is invariant to orthogonal basis changes, thereby helping capture robust intrinsic structures of the data. The empirical transfer performance of ALL-IN across new datasets with disparate features demonstrates its potential to mitigate the challenges posed by feature heterogeneity, contributing to the development of GFMs.

**Limitations and Future Work.** Current scalability for ALL-IN on extremely large graphs is constrained by its dense covariance operators; developing sparse approximations presents a key avenue for future research. Another promising direction involves exploring structured or learnable projections as alternatives to the random Gaussian projections.

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

# A   Related Work

**Graph Foundation Models (GFMs).** GFMs aim to learn representations that generalize across datasets and tasks, but achieving robust generalization remains challenging, especially when the node features change. Some recent approaches integrate large language models (LLMs) by converting graphs to text or embedding features through prompt-based designs [31, 53, 8, 12, 34, 7, 53, 20, 23, 40, 25, 52, 19, 38, 39], but this can lead to loss of structural details. Other recent works align feature spaces through projections [46, 49, 52, 11], perceiver-based encoders [27], computing analytical solutions (in the case of node classification) [54], encoding features into the graph structure [15, 16, 44] or encoding feature relationships [36]. While these methods advance GFM capabilities, they often require task-specific adaptations or face scalability issues, leaving a gap for truly input-space-agnostic solutions. ALL-IN offers a distinct path: it creates transferable representations by processing arbitrary input features through stochastic projections and node-covariance operators.

**Structural and Positional Encodings.** Concurrently, efforts to create universal graph representations include transferable structural and positional encodings (SPEs) [35, 3, 6, 25]. These SPEs aim to capture graph topology in a feature-agnostic manner, often within Graph Transformers or GNNs. While such SPEs can complement node features, ALL-IN directly addresses the challenge of heterogeneous node features themselves, transforming them into a robust, transferable format using their covariance structure, irrespective of any additional SPEs.

**Covariance networks.** Covariance matrices have also informed the design of neural networks. For instance, coVariance Neural Networks (VNNs) [37] process $d \times d$ sample covariance matrices, with $d$ the input feature dimension, which describe feature inter-correlations, offering benefits like stability to varying sample sizes and inspiring extensions for fairness [5] and sparsity [4]. Other related efforts focus on transferring principal components derived from data covariance [21]. While these methods analyze relationships between features using sample covariance matrices, ALL-IN constructs an $n \times n$ node-covariance matrix, with $n$ number of nodes. This operator quantifies similarities between pairs of nodes based on how their (randomly projected) features co-vary across dimensions. This distinct formulation is tailored to building transferable representations from graphs with heterogeneous node features, addressing a challenge different from that targeted by the aforementioned approaches.

**Generalization Theory of MPNNs.** Significant theoretical progress has advanced our understanding of generalization in Message Passing Neural Networks (MPNNs). As discussed in recent surveys [42, 50], these efforts often focus on how architectures and graph properties (such as maximum degree) influence the generalization gap, employing analytical tools like Rademacher complexity and PAC-Bayesian analysis to derive performance bounds [18, 30]. Other lines of work, leveraging concepts like covering numbers or graphon theory, investigate model stability and generalization under shifts in graph structure or topology, particularly in large-scale or evolving graph scenarios [29, 41]. While these foundational theories provide important insights into GNN expressivity and their ability to generalize, especially concerning structural variations, they typically assume a consistent definition of the input feature space across different graphs. The cross-dataset generalization challenge that ALL-IN addresses is distinct: we specifically tackle scenarios where graphs present node features from entirely different feature spaces, potentially varying in both the number of available features (dimensionality) and their semantic meaning between train (source) and test (target) graphs. Our theoretical framework (Appendix B) therefore focuses on establishing principles for robustness and transferability under such input feature space heterogeneity, aiming to complement existing generalization theories that predominantly address structural changes.

**Additional Efforts towards Graph Foundation Models.** Another significant challenge in graph transfer learning arises in settings like heterogeneous knowledge graphs, where models must generalize to unseen entities and relation types. Approaches such as ISDEA+ [17] and MTDEA [55] tackle this by employing set aggregation techniques over representations specific to edge types, aiming for equivariance to permutations of these types, supported by a "double equivariance" theoretical framework. Similarly, methods like InGram [28], ULTRA [16], TRIX [51], and MOTIF [24] construct explicit "relation graphs" to model interactions among different edge types. These works provide valuable solutions for structural and relational heterogeneity. In contrast, ALL-IN primarily addresses the distinct challenge of heterogeneity in input features, that is, varying feature dimensionalities and semantics across graphs. While the aforementioned methods focus on generalizing over graph schema and relation types (often assuming node features are not present), ALL-IN directly processes

arbitrary node features to derive transferable node-covariance operators and representations. Other efforts in graph representation learning aim for transferability across diverse graph tasks. For example, HoloGNN [1] proposes a framework to learn node representations that can be applied to various downstream tasks on a given graph or graphs. However, such approaches typically assume that the underlying node feature space remains consistent across these tasks. ALL-IN, conversely, is specifically designed to address the challenge of generalizing to new and unseen datasets where the node features themselves can differ fundamentally in dimensionality and semantics, a problem distinct from task-level transfer within a fixed feature domain.

# B    Theoretical Insights

This section establishes the theoretical foundations underpinning the ability of ALL-IN to handle heterogeneous input features and enable generalization across datasets. A core contribution is proving the method's robustness to common variations in feature representation. We first demonstrate that the computed node-covariance operators and the resulting node representations are invariant *in distribution* to arbitrary permutations of the input features, providing fundamental robustness to feature re-ordering. We then show that the *expected node-covariance operator* is invariant to general orthogonal transformations, ensuring robustness to the choice of orthonormal basis (Appendix B.1). Building on these properties, we validate the stochastic training procedure using Jensen's inequality under standard convexity assumptions (Appendix B.2). Finally, we discuss conditions supporting transferability, analyzing scenarios where the operator remains stable across graphs with differing feature distributions and proving its consistency for large projection dimensions (Appendix B.3). All proofs are provided in Appendix C.

## B.1    Invariance to Feature Space Transformations

A primary obstacle to cross-dataset transfer is the lack of feature standardization, leading to arbitrary differences in feature ordering and basis choice across datasets. Our approach, centered on node-covariance after random projection, inherently addresses these issues through invariance properties. First, the use of random isotropic Gaussian projections renders the process statistically insensitive to the order of input features. We formalize this by showing that the distribution of the projected feature matrix remains unchanged when the original features are permuted.

**Proposition B.1** (Distributional Invariance of Projected Features to Feature Permutation). *Let $X \in \mathbb{R}^{n \times d}$ be node features, $P \in \mathbb{R}^{d \times d}$ be any permutation matrix, and $h$ be the projection dimension. Let $C \in \mathbb{R}^{d \times h}$ be an isotropic Gaussian random matrix (i.e., $vec(C) \sim \mathcal{N}(\mathbf{0}, I_{dh})$). Define the projected features as $R^{(0)} = XC$ and the features projected after permutation as $\bar{R}^{(0)} = (XP)C$. Then $R^{(0)}$ and $\bar{R}^{(0)}$ are equal in distribution: $R^{(0)} \stackrel{d}{=} \bar{R}^{(0)}$.*

In essence, Proposition B.1 establishes that random projections effectively "mix" features, rendering their original ordering statistically irrelevant after projection. More importantly, the permutation invariance is characterized *in distribution*, rather than pointwise: for a fixed random projection $C$, the features in $R^{(0)}$ retain sensitivity to input permutations, thereby enabling a neural network to better capture the relationships between node features and topology.

To illustrate this concept, consider three nodes $u, v, w \in V$ with features $X_u = (0, 1)$, $X_v = (0, 1)$, and $X_w = (1, 0)$. Under strict (pointwise) permutation invariance, the embeddings of all nodes would be equivalent, obscuring the key distinction that $u$ and $v$ share identical features, whereas $w$ has a different feature. In contrast, distributional invariance ensures that the distributions of $R_u^{(0)}$, $R_v^{(0)}$, and $R_w^{(0)}$ are identical, yet individual forward passes yield different outcomes: given $C$, we have $R_u^{(0)} = R_v^{(0)} \neq R_w^{(0)}$. This property preserves the model's ability to distinguish between nodes $u$ and $v$ (which share the same features) and node $w$ (which has a different feature), while maintaining symmetry in the model's statistical behavior, thus striking a balance between permutation invariance and expressive power.

Next, we show that the NodeCov operators applied to the sequence $\{R^{(p)}\}_{p=0}^{k}$ yield features that are also distributionally invariant.

**Corollary B.2** (Distributional Invariance of Node Covariance Operators to Feature Permutation). *Let $X \in \mathbb{R}^{n \times d}$ be node features, and $P \in \mathbb{R}^{d \times d}$ be any permutation matrix. Let $R^{(0)} = XC$ be*

the initial projected features. Let $\mathcal{K} = \{\boldsymbol{K}^{(p)}\}_{p=0}^k$ be the set of node-covariance operators, where $\boldsymbol{K}^{(p)} = NodeCov(\boldsymbol{A}^p \boldsymbol{R}^{(0)})$ is computed using the deterministic function NodeCov, and $\boldsymbol{A}$ is the adjacency matrix. It follows directly from the distributional invariance of $\boldsymbol{R}^{(0)}$ that the entire set of operators $\mathcal{K}$ is also invariant in distribution to permutations of the input features $\boldsymbol{X}$. That is, if $\bar{\mathcal{K}}$ is the set of operators computed using $\boldsymbol{X}\boldsymbol{P}$ instead of $\boldsymbol{X}$, then $\mathcal{K} \stackrel{d}{=} \bar{\mathcal{K}}$.

The significance of Proposition B.1 and Corollary B.2 is substantial: it guarantees that the complete statistical behavior of $\boldsymbol{R}^{(0)}$ and the operators $\boldsymbol{K}^{(p)}$ central to ALL-IN is fundamentally robust to arbitrary input feature ordering, directly addressing a key source of heterogeneity across graph datasets. This distributional invariance also extends to the hidden representations $\boldsymbol{H}^{(\ell)}$, for all $\ell = 1 \ldots L$ derived from these operators, as shown in Theorem C.1 in Appendix C.

The stochastic projection matrix $\boldsymbol{C}$ plays a critical role beyond enabling the distributionally invariance properties discussed earlier; its use is intrinsically linked to the expressive capability of the learning framework. Training with node-covariance operators NodeCov($\boldsymbol{R}^{(0)}$) derived from these stochastic projections offers advantages over relying on a single, deterministically computed covariance operator, such as NodeCov($\boldsymbol{X}$). While NodeCov($\boldsymbol{X}$) provides a stable, pointwise feature-permutation invariant view of node similarities, it can obscure subtle but important distinctions between nodes. In contrast, individual stochastic realizations NodeCov($\boldsymbol{R}^{(0)}$) = NodeCov($\boldsymbol{X}\boldsymbol{C}$) (for a specific $\boldsymbol{C}$) can preserve these finer-grained distinctions, providing richer and more varied signals to the GNN. Theorem B.3 formalizes this concept by demonstrating that there exist instances where the stochastic operator NodeCov($\boldsymbol{X}\boldsymbol{C}$) can distinguish nodes that the deterministic operator NodeCov($\boldsymbol{X}$) cannot.

**Theorem B.3** (Distinguishability through $\boldsymbol{C}$). *There exist node features $\boldsymbol{X} \in \mathbb{R}^{n \times d}$, nodes $u, v \in V$ with $\boldsymbol{X}_u \neq \boldsymbol{X}_v$ such that NodeCov($\boldsymbol{X}$) makes $u$, $v$ indistinguishable (automorphic), but NodeCov($\boldsymbol{X}\boldsymbol{C}$) (for a.s. all $\boldsymbol{C}$) makes $u$, $v$ distinguishable (not automorphic).*

Finally, while distributional invariance covers permutations, analyzing the expected operator reveals broader robustness to basis changes and identifies the structure captured on average, as we show next.

**Theorem 2.1** (Expected Invariance to Orthogonal Transformations). *Let $\boldsymbol{X} \in \mathbb{R}^{n \times d}$ be node features, $\boldsymbol{Q} \in \mathbb{R}^{d \times d}$ be an orthogonal matrix, and $h$ be the projection dimension. Consider a random projection matrix $\boldsymbol{C} \in \mathbb{R}^{d \times h}$ with vec($\boldsymbol{C}$) $\sim \mathcal{N}(\boldsymbol{0}, \boldsymbol{I}_{dh})$. Let NodeCov($\boldsymbol{R}^{(0)}$) = $\frac{1}{h}(\boldsymbol{\Pi}_c \boldsymbol{R}^{(0)})(\boldsymbol{\Pi}_c \boldsymbol{R}^{(0)})^T$ be the Node Covariance operator (Equation (2)), where $\boldsymbol{\Pi}_c = \boldsymbol{I}_n - \frac{1}{n}\boldsymbol{1}_n\boldsymbol{1}_n^T$ is the centering matrix. Then, the expected Node Covariance computed from the stochastically projected features is invariant to the orthogonal transformation $\boldsymbol{Q}$:*

$$\mathbb{E}_{\boldsymbol{C}}[NodeCov(\boldsymbol{X}\boldsymbol{Q}\boldsymbol{C})] = \mathbb{E}_{\boldsymbol{C}}[NodeCov(\boldsymbol{X}\boldsymbol{C})] = \boldsymbol{\Pi}_c\boldsymbol{X}\boldsymbol{X}^T\boldsymbol{\Pi}_c \tag{5}$$

*where the expectation $\mathbb{E}_{\boldsymbol{C}}[\cdot]$ is over the random sampling of $\boldsymbol{C}$, and $\boldsymbol{\Pi}_c\boldsymbol{X}\boldsymbol{X}^T\boldsymbol{\Pi}_c$ is the Gram matrix of the centered original features.*

Theorem 2.1 demonstrates that the expected operator is agnostic to any choice of orthonormal basis (rotations, reflections, permutations) for the input features. Furthermore, identifying this stable expectation as the Gram matrix of centered original features ($\boldsymbol{\Pi}_c\boldsymbol{X}\boldsymbol{X}^T\boldsymbol{\Pi}_c$) reveals that ALL-IN, on average, recovers intrinsic, basis-invariant pairwise node similarities directly reflecting the original data structure, irrespective of the specific random projection used.

## B.2 Training Objective Upper Bound

ALL-IN computes the feature projection $\boldsymbol{R}^{(0)}$ and node-covariance operator $\boldsymbol{K}^{(0)} = NodeCov(\boldsymbol{X}\boldsymbol{C})$ using a stochastic projection matrix $\boldsymbol{C}$ sampled in each forward pass. We now validate this practical training approach by showing its connection to performance on the stable, expected final representation $\mathbb{E}_{\boldsymbol{C}}[\boldsymbol{H}^{(L)}]$, assuming common convexity conditions for the final prediction layer.

**Theorem B.4** (Loss Upper Bound). *Let $\boldsymbol{H}^{(L)} \in \mathbb{R}^{n \times h^{(L)}}$ be the final node representations computed by ALL-IN, dependent on the initial random projection $\boldsymbol{C}$. Let $\phi : \mathbb{R}^{n \times h^{(L)}} \to \mathbb{R}^{n \times t}$ be the final prediction layer, and let $\mathcal{L}(\cdot, \boldsymbol{Y})$ be the loss function comparing predictions to ground truth labels $\boldsymbol{Y}$. Assume that the composite function $f(\boldsymbol{H}^{(L)}) = \mathcal{L}(\phi(\boldsymbol{H}^{(L)}), \boldsymbol{Y})$ is convex with respect to the final node representations $\boldsymbol{H}^{(L)}$ (this holds, for instance, if $\phi$ is a linear map or linear plus softmax, and*

$\mathcal{L}$ is cross-entropy or mean squared error). Then, our stochastic optimization objective provides an upper bound for the loss of the expected representation:

$$\underbrace{\mathcal{L}(\phi(\mathbb{E}_{\boldsymbol{C}}[\boldsymbol{H}^{(L)}]), \boldsymbol{Y})}_{\text{Loss of Expected Representation}} \leq \underbrace{\mathbb{E}_{\boldsymbol{C}}[\mathcal{L}(\phi(\boldsymbol{H}^{(L)}), \boldsymbol{Y})]}_{\text{Expected Loss (Training Objective)}} \tag{6}$$

where the expectation $\mathbb{E}_{\boldsymbol{C}}[\cdot]$ is taken over the random projection matrix $\boldsymbol{C}$.

Theorem B.4 provides key theoretical support for training with stochastic projections. The inequality (Equation (6)) establishes that the expected loss minimized during training (RHS) serves as a guaranteed upper bound for the loss evaluated on the stable, expected final representation (LHS). Therefore, minimizing the empirical average loss (approximating the RHS) acts as a theoretically sound surrogate objective, implicitly promoting minimization of the loss associated with the expected representation, thus validating our stochastic approach.

## B.3 Conditions for Transferability and Operator Consistency

Beyond invariance, achieving transfer across graphs with fundamentally different feature distributions ($\boldsymbol{X}^{(1)}, \boldsymbol{X}^{(2)}$ for graphs $G_1, G_2$) relies on the stability of the underlying structure captured by the expected operator, $\mathbb{E}_{\boldsymbol{C}}[\boldsymbol{K}^{(0)}] = \Pi_c \boldsymbol{X} \boldsymbol{X}^T \Pi_c$. We posit that such stability can arise when graphs share intrinsic properties. Plausible scenarios where such stability in the expected operator might arise include graphs exhibiting similar relational structures tied to node features (e.g., comparable label homophily if features reflect labels), originating from a shared underlying generative process (e.g., common SBM or graphon influencing features), or possessing similar distributions of node roles (e.g., hubs, bridges) if features are role-informative. In these cases, even if the specific feature realizations differ, the resulting $\Pi_c \boldsymbol{X}^{(i)} (\boldsymbol{X}^{(i)})^T \Pi_c$ matrices may capture analogous relational structures.

For this potential transfer to be practically realized, the stochastic operator $\boldsymbol{K}_h^{(0)}$ computed using a finite projection dimension $h$ must reliably estimate its expectation. This holds for large $h$.

**Proposition B.5** (Consistency of Projected Node Covariance). *Let $\boldsymbol{X} \in \mathbb{R}^{n \times d}$ be node features. For a projection dimension $h$, let $\boldsymbol{C} \in \mathbb{R}^{d \times h}$ be such that $vec(\boldsymbol{C}) \sim \mathcal{N}(\boldsymbol{0}, \boldsymbol{I}_{dh})$. Define the stochastic node-covariance operator $\boldsymbol{K}_h^{(0)} = NodeCov(\boldsymbol{X}\boldsymbol{C}) = \frac{1}{h}(\Pi_c \boldsymbol{X}\boldsymbol{C})(\Pi_c \boldsymbol{X}\boldsymbol{C})^T$, where $\Pi_c$ is the centering matrix. Then, $\boldsymbol{K}_h^{(0)}$ converges in probability to its expected value as $h \to \infty$:*

$$\boldsymbol{K}_h^{(0)} \xrightarrow{p} \mathbb{E}_{\boldsymbol{C}}[\boldsymbol{K}_h^{(0)}] = \Pi_c \boldsymbol{X} \boldsymbol{X}^T \Pi_c \quad as\ h \to \infty. \tag{7}$$

This consistency connects theory to practice. It shows that for a sufficiently large $h$, the operator accurately reflects the stable expected operator $\Pi_c \boldsymbol{X} \boldsymbol{X}^T \Pi_c$. Therefore, if two graphs have aligned expected operators (due to shared properties), using a large enough $h$ allows ALL-IN to effectively leverage these shared underlying structures, facilitating transfer across disparate feature spaces.

# C Additional Theoretical Considerations and Proofs

**Proposition B.1** (Distributional Invariance of Projected Features to Feature Permutation). *Let $\boldsymbol{X} \in \mathbb{R}^{n \times d}$ be node features, $\boldsymbol{P} \in \mathbb{R}^{d \times d}$ be any permutation matrix, and $h$ be the projection dimension. Let $\boldsymbol{C} \in \mathbb{R}^{d \times h}$ be an isotropic Gaussian random matrix (i.e., $vec(\boldsymbol{C}) \sim \mathcal{N}(\boldsymbol{0}, \boldsymbol{I}_{dh})$). Define the projected features as $\boldsymbol{R}^{(0)} = \boldsymbol{X}\boldsymbol{C}$ and the features projected after permutation as $\bar{\boldsymbol{R}}^{(0)} = (\boldsymbol{X}\boldsymbol{P})\boldsymbol{C}$. Then $\boldsymbol{R}^{(0)}$ and $\bar{\boldsymbol{R}}^{(0)}$ are equal in distribution: $\boldsymbol{R}^{(0)} \stackrel{d}{=} \bar{\boldsymbol{R}}^{(0)}$.*

*Proof.* Let $\boldsymbol{C}$ have columns $\boldsymbol{c}_1, \ldots, \boldsymbol{c}_h$. Since the entries $C_{ik}$ are i.i.d $\mathcal{N}(0, 1)$, each column $\boldsymbol{c}_j \sim \mathcal{N}(\boldsymbol{0}, \boldsymbol{I}_d)$ and the columns are mutually independent.

Consider the matrix $\bar{\boldsymbol{C}} = \boldsymbol{P}^T \boldsymbol{C}$. Since $\boldsymbol{P}$ is a permutation matrix, $\boldsymbol{P}^T$ is also a permutation matrix and is orthogonal, that is $\boldsymbol{P}^T (\boldsymbol{P}^T)^T = \boldsymbol{P}^T \boldsymbol{P} = \boldsymbol{I}_d$.

The columns of $\bar{\boldsymbol{C}}$ are $\bar{\boldsymbol{c}}_j = \boldsymbol{P}^T \boldsymbol{c}_j$. Since $\boldsymbol{c}_j \sim \mathcal{N}(\boldsymbol{0}, \boldsymbol{I}_d)$ and $\boldsymbol{P}^T$ is orthogonal, then

$$\bar{\boldsymbol{c}}_j \sim \mathcal{N}(\boldsymbol{P}^T \boldsymbol{0}, \boldsymbol{P}^T \boldsymbol{I}_d (\boldsymbol{P}^T)^T) = \mathcal{N}(\boldsymbol{0}, \boldsymbol{P}^T \boldsymbol{P}) = \mathcal{N}(\boldsymbol{0}, \boldsymbol{I}_d) \tag{8}$$

Furthermore, since $c_1, \ldots, c_h$ are independent, the transformed columns $\bar{c}_1, \ldots, \bar{c}_h$ are also independent. Thus, the matrix $\bar{C}$ has the same distribution as $C$, i.e., $\bar{C} \overset{d}{=} C$.

Now consider $\bar{R}^{(0)} = (XP)C$. Since $C \overset{d}{=} \bar{C}$, we can write:

$$\bar{R}^{(0)} \overset{d}{=} (XP)\bar{C}$$

Substitute $\bar{C} = P^T C$:

$$\bar{R}^{(0)} \overset{d}{=} (XP)(P^T C) = X(PP^T)C$$

Since $P$ is orthogonal, $PP^T = I_d$.

$$\bar{R}^{(0)} \overset{d}{=} XI_d C = XC = R$$

Thus, $R$ and $\bar{R}^{(0)}$ are equal in distribution. $\qquad\square$

**Corollary B.2** (Distributional Invariance of Node Covariance Operators to Feature Permutation)**.** *Let $X \in \mathbb{R}^{n \times d}$ be node features, and $P \in \mathbb{R}^{d \times d}$ be any permutation matrix. Let $R^{(0)} = XC$ be the initial projected features. Let $\mathcal{K} = \{K^{(p)}\}_{p=0}^{k}$ be the set of node-covariance operators, where $K^{(p)} = \text{NodeCov}(A^p R^{(0)})$ is computed using the deterministic function NodeCov, and $A$ is the adjacency matrix. It follows directly from the distributional invariance of $R^{(0)}$ that the entire set of operators $\mathcal{K}$ is also invariant in distribution to permutations of the input features $X$. That is, if $\bar{\mathcal{K}}$ is the set of operators computed using $XP$ instead of $X$, then $\mathcal{K} \overset{d}{=} \bar{\mathcal{K}}$.*

*Proof.* Let $g_p(R^{(0)}) = \text{NodeCov}(A^p R^{(0)})$ be the deterministic function that computes the p-th order node covariance operator from the initial projected features $R^{(0)}$. From Proposition B.1, we have $R^{(0)} \overset{d}{=} \bar{R}^{(0)}$. Since applying a deterministic function $g_p$ to random variables that are equal in distribution results in outputs that are equal in distribution, we have $g_p(R^{(0)}) \overset{d}{=} g_p(\bar{R}^{(0)})$, which means $K^{(p)} \overset{d}{=} \bar{K}^{(p)}$ for each $p = 0 \ldots k$. Furthermore, since all operators $K^{(p)}$ in $\mathcal{K}$ are derived from the same $R^{(0)}$, and all operators $\bar{K}^{(p)}$ in $\bar{\mathcal{K}}$ are derived from $\bar{R}^{(0)}$, the distributional equality extends to the joint distribution of the sets: $\mathcal{K} \overset{d}{=} \bar{\mathcal{K}}$. $\qquad\square$

**Theorem C.1** (Distributional Invariance of Hidden Representations to Input Permutation)**.** *Let $X \in \mathbb{R}^{n \times d}$ be node features, and $P \in \mathbb{R}^{d \times d}$ be any permutation matrix. Let $R^{(0)} = XC$ be the initial projected features, and $\mathcal{K} = \{K^{(p)}\}_{p=0}^{k}$ be the set of node-covariance operators. Let the initial hidden representation be $H^{(0)} = R^{(0)} \oplus S$, where $S$ is a structural encoding matrix independent of $X$. Subsequent hidden representations $H^{(\ell)}$ for $\ell = 1, \ldots, L$ are computed by a deterministic GNN layer function.*

*The initial hidden representation $H^{(0)}$ and all subsequent hidden representations $H^{(\ell)}$ for $\ell = 1, \ldots, L$ are invariant in distribution to permutations of the input features $X$. That is, if $\bar{H}^{(\ell)}$ are the representations computed using $XP$ instead of $X$, then $H^{(\ell)} \overset{d}{=} \bar{H}^{(\ell)}$ for all $\ell$.*

*Proof.* We proceed by induction on the layer index $\ell$.

**Base Case ($\ell = 0$).** Let $R^{(0)} = XC$ and $\bar{R}^{(0)} = (XP)C$. The initial hidden representations are $H^{(0)} = R^{(0)} \oplus S$ and $\bar{H}^{(0)} = \bar{R}^{(0)} \oplus S$. From Proposition B.1, we know that $R^{(0)} \overset{d}{=} \bar{R}^{(0)}$. Since the structural encoding $S$ is assumed independent of $X$ (and thus fixed with respect to the permutation $P$), and the concatenation operation $\oplus$ is a deterministic function, applying this function preserves the distributional equality. Therefore, $H^{(0)} = R^{(0)} \oplus S \overset{d}{=} \bar{R}^{(0)} \oplus S = \bar{H}^{(0)}$. The base case holds.

**Inductive Hypothesis.** Assume that for some layer $\ell - 1 \geq 0$, the hidden representations are equal in distribution: $H^{(\ell-1)} \overset{d}{=} \bar{H}^{(\ell-1)}$.

**Inductive Step (Layer $\ell$).** The hidden representations at layer $\ell$ are computed as:

$$H^{(\ell)} = F_\ell(H^{(\ell-1)}, \mathcal{O})$$

$$\bar{H}^{(\ell)} = F_\ell(\bar{H}^{(\ell-1)}, \bar{\mathcal{O}})$$

where $F_\ell$ represents the deterministic computation performed by the $\ell$-th GNN layer (given fixed learned weights), $\mathcal{O} = \{I, A\} \cup \mathcal{K}$ with $\mathcal{K} = \{\text{NodeCov}(A^p R^{(0)})\}_{p=0}^k$, and $\bar{\mathcal{O}} = \{I, A\} \cup \bar{\mathcal{K}}$ with $\bar{\mathcal{K}} = \{\text{NodeCov}(A^p \bar{R}^{(0)})\}_{p=0}^k$.

From Corollary B.2, we know that the set of random operators $\mathcal{K}$ is equal in distribution to $\bar{\mathcal{K}}$, i.e., $\mathcal{K} \overset{d}{=} \bar{\mathcal{K}}$. Since $I$ and $A$ are fixed, the full set of operators used by the layer also satisfies $\mathcal{O} \overset{d}{=} \bar{\mathcal{O}}$.

Now consider the inputs to the function $F_\ell$. The pair $(H^{(\ell-1)}, \mathcal{O})$ determines $H^{(\ell)}$, and the pair $(\bar{H}^{(\ell-1)}, \bar{\mathcal{O}})$ determines $\bar{H}^{(\ell)}$. Both $H^{(\ell-1)}$ and $\mathcal{O}$ are deterministic functions of the initial projection $R^{(0)}$ (and fixed elements $S, A, I$, and layer weights). Let $J$ be the function representing the computation up to layer $\ell - 1$ and the computation of operators, such that $(H^{(\ell-1)}, \mathcal{O}) = J(R^{(0)}, S, A, I, \text{Weights})$ Similarly, $(\bar{H}^{(\ell-1)}, \bar{\mathcal{O}}) = J(\bar{R}^{(0)}, S, A, I, \text{Weights})$.

Since $R^{(0)} \overset{d}{=} \bar{R}^{(0)}$ (Proposition B.1) and $J$ is a deterministic function, it follows that the joint distribution of the outputs is preserved:

$$(H^{(\ell-1)}, \mathcal{O}) \overset{d}{=} (\bar{H}^{(\ell-1)}, \bar{\mathcal{O}})$$

This establishes that the inputs to the deterministic layer function $F_\ell$ are equal in distribution. Applying the deterministic function $F_\ell$ preserves this equality:

$$H^{(\ell)} = F_\ell(H^{(\ell-1)}, \mathcal{O}) \overset{d}{=} F_\ell(\bar{H}^{(\ell-1)}, \bar{\mathcal{O}}) = \bar{H}^{(\ell)}$$

Thus, the inductive step holds. $\qquad\square$

**Theorem B.3** (Distinguishability through $C$). *There exist node features $X \in \mathbb{R}^{n \times d}$, nodes $u, v \in V$ with $X_u \neq X_v$ such that NodeCov$(X)$ makes $u, v$ indistinguishable (automorphic), but NodeCov$(XC)$ (for a.s. all $C$) makes $u, v$ distinguishable (not automorphic).*

*Proof.* We will show that there exists $X$, $u$, $v$ such that (1) nodes $u$ and $v$ are automorphic within NodeCov$(X)$, and consequently, the GNN, when using NodeCov$(X)$ as the operator and identical initial embeddings, produces identical final representations for these nodes. (2) For the same $X$, with probability 1 (over the draw of $C$), nodes $u$ and $v$ are **not** automorphic and therefore distinguishable in NodeCov$(XC)$. We provide a constructive example. Let $n = 3$ nodes $\{u, v, w\}$ and $d = 3$ features. Consider the feature matrix $X$:

$$X = \begin{pmatrix} X_u^T \\ X_v^T \\ X_w^T \end{pmatrix} = \begin{pmatrix} 1 & 0 & 1 \\ 0 & 1 & 1 \\ 1 & 1 & 0 \end{pmatrix}$$

Here, $X_u = (1, 0, 1)^T$, $X_v = (0, 1, 1)^T$, and $X_w = (1, 1, 0)^T$. Clearly, $X_u \neq X_v$.

**Proof for item (1).** The column means of $X$ are $\bar{X}_{\text{col}} = (2/3, 2/3, 2/3)^T$. The centered feature matrix $X_c = \Pi_c X = X - 1_3 \bar{X}_{\text{col}}^T$ is:

$$X_c = \begin{pmatrix} 1/3 & -2/3 & 1/3 \\ -2/3 & 1/3 & 1/3 \\ 1/3 & 1/3 & -2/3 \end{pmatrix}$$

Then

$$\text{NodeCov}(X) = \begin{pmatrix} 2/9 & -1/9 & -1/9 \\ -1/9 & 2/9 & -1/9 \\ -1/9 & -1/9 & 2/9 \end{pmatrix}.$$

In the weighted graph defined by NodeCov$(X)$, all nodes are automorphic to each other. If a GNN uses NodeCov$(\bar{X})$ as its feature-derived operator and starts with identical initial embeddings for all nodes, standard message passing layers will preserve this symmetry, leading to identical final representations $H_u^{(L)} = H_v^{(L)} = H_w^{(L)}$. Thus, such a GNN cannot distinguish $u$ from $v$.

**Proof for item (2).** Let $R^{(0)} = XC$. The rows of $R^{(0)}$ are $R_u^{(0)} = X_u^T C$, $R_v^{(0)} = X_v^T C$, $R_w^{(0)} = X_w^T C$. Since $X_u \neq X_v$ and $C$ is drawn from a continuous distribution (Gaussian entries), $X_u^T C \neq$

$X_v^T C$ with probability 1. Thus, $R_u^{(0)} \neq R_v^{(0)}$ almost surely. Let $R_c^{(0)} = \Pi_c R^{(0)}$. The rows of $R_c$ are $R_{c,u}^{(0)}, R_{c,v}^{(0)}, R_{c,w}^{(0)}$. Since $R_u^{(0)} \neq R_v^{(0)}$, it follows that $R_{c,u}^{(0)} \neq R_{c,v}^{(0)}$ almost surely (unless $\Pi_c$ projects their difference to zero, which is a measure zero event for a fixed $X$ and random $C$). The operator is $K^{(0)} = \text{NodeConv}(XC) = \frac{1}{h} R_c R_c^T$. An element $(K^{(0)})_{ij} = \frac{1}{h} R_{c,i}^{(0)} \cdot R_{c,j}^{(0)}$. Consider the specific symmetry that existed for $\text{NodeConv}(X)$, e.g., $(\text{NodeConv}(X))_{uw} = (\text{NodeConv}(X))_{vw} = -1/9$. For $K^{(0)}$, we compare $(K^{(0)})_{uw} = \frac{1}{h} R_{c,u}^{(0)} \cdot R_{c,w}^{(0)}$ and $(K^{(0)})_{vw} = \frac{1}{h} R_{c,v}^{(0)} \cdot R_{c,w}^{(0)}$. These are equal if $(R_{c,u}^{(0)} - R_{c,v}^{(0)}) \cdot R_{c,w}^{(0)} = 0$. Since $R_{c,u}^{(0)} - R_{c,v}^{(0)} \neq \mathbf{0}$ almost surely, and $R_{c,w}^{(0)}$ is a random vector (whose distribution depends on $C$), the event that their dot product is exactly zero has probability 0 for continuous distributions unless one of them is deterministically zero (which is not the case here a.s.). Therefore, with probability 1, $(K^{(0)})_{uw} \neq (K^{(0)})_{vw}$. This breaks the specific symmetry that made node u and node v have equivalent relational profiles to node w in $\text{NodeCov}(X)$. More generally, the matrix $K^{(0)}$ will not, with probability 1, exhibit the high degree of symmetry found in $\text{NodeCov}(X)$ for this specific $X$. Thus, nodes $u$ and $v$ will generally not be automorphic with respect to $K^{(0)}$ in the same way they were for $\text{NodeCov}(X)$. A GNN using this specific realization $K^{(0)}$ (and identical initial embeddings, can now potentially produce $H_u^{(L)} \neq H_v^{(L)}$ because the operator $K^{(0)}$ provides different relational information for $u$ and $v$.

$\square$

**Theorem 2.1** (Expected Invariance to Orthogonal Transformations). *Let $X \in \mathbb{R}^{n \times d}$ be node features, $Q \in \mathbb{R}^{d \times d}$ be an orthogonal matrix, and h be the projection dimension. Consider a random projection matrix $C \in \mathbb{R}^{d \times h}$ with $vec(C) \sim \mathcal{N}(\mathbf{0}, I_{dh})$. Let $NodeCov(R^{(0)}) = \frac{1}{h}(\Pi_c R^{(0)})(\Pi_c R^{(0)})^T$ be the Node Covariance operator (Equation* (2)*), where $\Pi_c = I_n - \frac{1}{n}\mathbf{1}_n \mathbf{1}_n^T$ is the centering matrix. Then, the expected Node Covariance computed from the stochastically projected features is invariant to the orthogonal transformation $Q$:*

$$\mathbb{E}_C[NodeCov(XQC)] = \mathbb{E}_C[NodeCov(XC)] = \Pi_c X X^T \Pi_c \tag{5}$$

*where the expectation $\mathbb{E}_C[\cdot]$ is over the random sampling of $C$, and $\Pi_c X X^T \Pi_c$ is the Gram matrix of the centered original features.*

*Proof.* Let $R^{(0)} = XC$. Using the definition of the NodeCov operator and properties of the centering matrix $\Pi_c$:

$$\text{NodeCov}(R^{(0)}) = \frac{1}{h}(\Pi_c R^{(0)})(\Pi_c R^{(0)})^T$$

$$= \frac{1}{h}\Pi_c(XC)(XC)^T\Pi_c^T$$

$$= \frac{1}{h}\Pi_c X C C^T X^T \Pi_c$$

Taking the expectation over $C$:

$$\mathbb{E}_C[\text{NodeCov}(XC)] = \mathbb{E}_C\left[\frac{1}{h}\Pi_c X C C^T X^T \Pi_c\right]$$

$$= \frac{1}{h}\Pi_c X \mathbb{E}_C[CC^T] X^T \Pi_c \quad \text{(by linearity of expectation)}$$

We evaluate $\mathbb{E}_C[CC^T]$. Let $c_j \in \mathbb{R}^d$ be the j-th column of $C$. Since the entries of $C$ are i.i.d. $\mathcal{N}(0,1)$, each column vector $c_j$ follows $c_j \sim \mathcal{N}(\mathbf{0}, I_d)$. Therefore, $E[c_j c_j^T] = I_d$. Using linearity of expectation:

$$\mathbb{E}_C[CC^T] = \mathbb{E}_C\left[\sum_{j=1}^{h} c_j c_j^T\right] = \sum_{j=1}^{h} \mathbb{E}_C[c_j c_j^T] = \sum_{j=1}^{h} I_d = h I_d$$

Substituting this back:

$$\mathbb{E}_C[\text{NodeCov}(XC)] = \frac{1}{h}\Pi_c X (h I_d) X^T \Pi_c = \Pi_c X X^T \Pi_c$$

Now consider the transformed features $\bar{X} = XQ$. Let $\bar{R}^{(0)} = \bar{X}C = XQC$. We compute $\mathbb{E}_C[\text{NodeCov}(\bar{R}^{(0)})]$:

$$\text{NodeCov}(\bar{R}^{(0)}) = \frac{1}{h}(\Pi_c\bar{R}^{(0)})(\Pi_c\bar{R}^{(0)})^T$$

$$= \frac{1}{h}\Pi_c(XQC)(XQC)^T\Pi_c$$

$$= \frac{1}{h}\Pi_c XQCC^TQ^TX^T\Pi_c$$

Taking the expectation over $C$:

$$\mathbb{E}_C[\text{NodeCov}(XQC)] = \frac{1}{h}\Pi_c XQ\mathbb{E}_C[CC^T]Q^TX^T\Pi_c$$

$$= \frac{1}{h}\Pi_c XQ(hI_d)Q^TX^T\Pi_c \quad \text{(using } E[CC^T] = hI_d)$$

$$= \Pi_c XQI_dQ^TX^T\Pi_c$$

$$= \Pi_c X(QQ^T)X^T\Pi_c$$

$$= \Pi_c XI_dX^T\Pi_c \quad \text{(since } Q \text{ is orthogonal, } QQ^T = I_d)$$

$$= \Pi_c XX^T\Pi_c$$

Thus, $\mathbb{E}_C[\text{NodeCov}(XQC)] = \mathbb{E}_C[\text{NodeCov}(XC)] = \Pi_c XX^T\Pi_c$. $\qquad \square$

**Theorem B.4** (Loss Upper Bound). *Let $H^{(L)} \in \mathbb{R}^{n\times h^{(L)}}$ be the final node representations computed by* ALL-IN*, dependent on the initial random projection $C$. Let $\phi : \mathbb{R}^{n\times h^{(L)}} \to \mathbb{R}^{n\times t}$ be the final prediction layer, and let $\mathcal{L}(\cdot, Y)$ be the loss function comparing predictions to ground truth labels $Y$. Assume that the composite function $f(H^{(L)}) = \mathcal{L}(\phi(H^{(L)}), Y)$ is convex with respect to the final node representations $H^{(L)}$ (this holds, for instance, if $\phi$ is a linear map or linear plus softmax, and $\mathcal{L}$ is cross-entropy or mean squared error). Then, our stochastic optimization objective provides an upper bound for the loss of the expected representation:*

$$\underbrace{\mathcal{L}(\phi(\mathbb{E}_C[H^{(L)}]), Y)}_{\text{Loss of Expected Representation}} \leq \underbrace{\mathbb{E}_C[\mathcal{L}(\phi(H^{(L)}), Y)]}_{\text{Expected Loss (Training Objective)}} \qquad (6)$$

*where the expectation $\mathbb{E}_C[\cdot]$ is taken over the random projection matrix $C$.*

*Proof.* The proof follows directly from Jensen's inequality for vector- or matrix-valued random variables.

Let the random variable be the final hidden representation $Z = H^{(L)}$, which is a function of the random projection matrix $C$.

By assumption, the function $f$ is convex with respect to its input argument $H^{(L)}$. Jensen's inequality states that for a convex function $f$ and a random variable $Z$ with finite expectation, $f(\mathbb{E}[Z]) \leq \mathbb{E}[f(Z)]$. Applying this with $Z = H^{(L)}$ and the defined function $f$, we get:

$$\mathcal{L}(\phi(\mathbb{E}_C[H^{(L)}]), Y) \leq \mathbb{E}_C[\mathcal{L}(\phi(H^{(L)}), Y)]$$

which is the desired result. $\qquad \square$

**Proposition B.5** (Consistency of Projected Node Covariance). *Let $X \in \mathbb{R}^{n\times d}$ be node features. For a projection dimension $h$, let $C \in \mathbb{R}^{d\times h}$ be such that $vec(C) \sim \mathcal{N}(0, I_{dh})$. Define the stochastic node-covariance operator $K_h^{(0)} = \text{NodeCov}(XC) = \frac{1}{h}(\Pi_c XC)(\Pi_c XC)^T$, where $\Pi_c$ is the centering matrix. Then, $K_h^{(0)}$ converges in probability to its expected value as $h \to \infty$:*

$$K_h^{(0)} \xrightarrow{p} \mathbb{E}_C[K_h^{(0)}] = \Pi_c XX^T\Pi_c \quad \text{as } h \to \infty. \qquad (7)$$

*Proof.* Let $C = [c_1, \ldots, c_h]$ denote the random projection matrix, where each column $c_j \in \mathbb{R}^d$ is a random vector. Since the entries of $C$ are sampled i.i.d. from $\mathcal{N}(0, 1)$, the columns $c_j$ are independent and identically distributed according to $c_j \sim \mathcal{N}(0, I_d)$.

The stochastic node-covariance operator $\boldsymbol{K}_h^{(0)}$ (Equation (2)) can be rewritten as:

$$
\begin{aligned}
\boldsymbol{K}_h^{(0)} &= \frac{1}{h}(\boldsymbol{\Pi}_c \boldsymbol{X} \boldsymbol{C})(\boldsymbol{\Pi}_c \boldsymbol{X} \boldsymbol{C})^T \\
&= \frac{1}{h}(\boldsymbol{\Pi}_c \boldsymbol{X}[\boldsymbol{c}_1, \ldots, \boldsymbol{c}_h])(\boldsymbol{\Pi}_c \boldsymbol{X}[\boldsymbol{c}_1, \ldots, \boldsymbol{c}_h])^T \\
&= \frac{1}{h}([\boldsymbol{\Pi}_c \boldsymbol{X} \boldsymbol{c}_1, \ldots, \boldsymbol{\Pi}_c \boldsymbol{X} \boldsymbol{c}_h])([\boldsymbol{\Pi}_c \boldsymbol{X} \boldsymbol{c}_1, \ldots, \boldsymbol{\Pi}_c \boldsymbol{X} \boldsymbol{c}_h])^T \\
&= \frac{1}{h}\sum_{j=1}^{h}(\boldsymbol{\Pi}_c \boldsymbol{X} \boldsymbol{c}_j)(\boldsymbol{\Pi}_c \boldsymbol{X} \boldsymbol{c}_j)^T \quad \text{(using block matrix multiplication definition)}
\end{aligned}
$$

Let us define the random matrix $\boldsymbol{Y}_j \in \mathbb{R}^{n \times n}$ as:

$$
\boldsymbol{Y}_j = (\boldsymbol{\Pi}_c \boldsymbol{X} \boldsymbol{c}_j)(\boldsymbol{\Pi}_c \boldsymbol{X} \boldsymbol{c}_j)^T
$$

Since the columns $\boldsymbol{c}_j$ are i.i.d. and $\boldsymbol{Y}_j$ is a fixed function of $\boldsymbol{c}_j$ (given the fixed matrices $\boldsymbol{X}$ and $\boldsymbol{\Pi}_c$), the random matrices $\boldsymbol{Y}_1, \boldsymbol{Y}_2, \ldots, \boldsymbol{Y}_h$ are also independent and identically distributed (i.i.d.).

The operator $\boldsymbol{K}_h^{(0)}$ can thus be written as the sample mean of these i.i.d. random matrices:

$$
\boldsymbol{K}_h^{(0)} = \frac{1}{h}\sum_{j=1}^{h}\boldsymbol{Y}_j
$$

Now, we compute the expected value of $\boldsymbol{Y}_j$. Using the linearity of expectation and the property that $\boldsymbol{\Pi}_c$ and $\boldsymbol{X}$ are constant with respect to the expectation over $\boldsymbol{C}$ (and $\boldsymbol{\Pi}_c = \boldsymbol{\Pi}_c^T$):

$$
\begin{aligned}
\mathbb{E}[\boldsymbol{Y}_j] &= \mathbb{E}[(\boldsymbol{\Pi}_c \boldsymbol{X} \boldsymbol{c}_j)(\boldsymbol{\Pi}_c \boldsymbol{X} \boldsymbol{c}_j)^T] \\
&= \mathbb{E}[\boldsymbol{\Pi}_c \boldsymbol{X} \boldsymbol{c}_j \boldsymbol{c}_j^T \boldsymbol{X}^T \boldsymbol{\Pi}_c^T] \\
&= \boldsymbol{\Pi}_c \boldsymbol{X} \mathbb{E}[\boldsymbol{c}_j \boldsymbol{c}_j^T] \boldsymbol{X}^T \boldsymbol{\Pi}_c
\end{aligned}
$$

Since $\boldsymbol{c}_j \sim \mathcal{N}(\boldsymbol{0}, \boldsymbol{I}_d)$, we know that $\mathbb{E}[\boldsymbol{c}_j \boldsymbol{c}_j^T] = \text{Cov}(\boldsymbol{c}_j) + \mathbb{E}[\boldsymbol{c}_j]\mathbb{E}[\boldsymbol{c}_j]^T = \boldsymbol{I}_d + \boldsymbol{0}\boldsymbol{0}^T = \boldsymbol{I}_d$. Substituting this in:

$$
\mathbb{E}[\boldsymbol{Y}_j] = \boldsymbol{\Pi}_c \boldsymbol{X} \boldsymbol{I}_d \boldsymbol{X}^T \boldsymbol{\Pi}_c = \boldsymbol{\Pi}_c \boldsymbol{X} \boldsymbol{X}^T \boldsymbol{\Pi}_c
$$

Let $\boldsymbol{K}_{\exp} = \boldsymbol{\Pi}_c \boldsymbol{X} \boldsymbol{X}^T \boldsymbol{\Pi}_c$. We have shown that $\mathbb{E}[\boldsymbol{Y}_j] = \boldsymbol{K}_{\exp}$. Since $\boldsymbol{X}$ is a fixed finite matrix, and the moments of Gaussian variables are finite, the expectation $\mathbb{E}[\boldsymbol{Y}_j]$ exists and is finite.

We have $\boldsymbol{K}_h^{(0)}$ as the sample mean of $h$ i.i.d. random matrices $\boldsymbol{Y}_j$, each with finite expectation $\boldsymbol{K}_{\exp}$. By the Weak Law of Large Numbers, applicable to sums of i.i.d. random vectors or matrices (considering convergence element-wise or in matrix norm), the sample mean converges in probability to the expected value as the number of samples $h$ goes to infinity. Therefore, for each entry $(a, b)$ of the matrices:

$$
(\boldsymbol{K}_h^{(0)})_{ab} = \frac{1}{h}\sum_{j=1}^{h}(\boldsymbol{Y}_j)_{ab} \xrightarrow{p} \mathbb{E}[(\boldsymbol{Y}_j)_{ab}] = (\boldsymbol{K}_{\exp})_{ab} \quad \text{as } h \to \infty
$$

This element-wise convergence implies convergence in probability for the matrix:

$$
\boldsymbol{K}_h^{(0)} \xrightarrow{p} \boldsymbol{K}_{\exp} = \boldsymbol{\Pi}_c \boldsymbol{X} \boldsymbol{X}^T \boldsymbol{\Pi}_c \quad \text{as } h \to \infty.
$$

This completes the proof. $\qquad\square$

## D  Additional Results

### D.1  Categorization and Description of Baselines

Table 2 compares our approach against diverse families of baselines evaluated on node classification benchmarks. We group methods into four primary categories: (i) SUPERVISED GNNS that are trained

from scratch on each dataset, (ii) LLM-AUGMENTED GNNs where the node features are enhanced using language models, (iii) LLM-BASED REASONING that convert the graph into a compatible input to pre-trained LLMs, and (iv) GNN-BASED methods.

**SUPERVISED BASELINES** include (a) MLP: a multi-layer perceptron directly on the target dataset features without using graph structure; serves as a non-graph baseline. (b) GCN [26]: trained from scratch on the target dataset (c) GIN [47] trained from scratch, included to represent expressive message-passing GNNs in supervised settings. These fall under supervised baselines as they do not perform pretraining or transfer, and rely solely on training from scratch on each dataset.

**LLM-AUGMENTED GNNs** include (a) OFA [31]: constructs a prompt-augmented graph using text nodes and pretrains an RGCN to enable in-context transfer across node/link/graph tasks; falls here for fusing text prompts with GNN structure and relying on LLM embeddings. (b) GLEM-LM [8]: Enhances GNNs using sentence-level text embeddings from a frozen LLM; categorized here due to its augmentation of GNN input via LLM-derived features. These are classified as LLM-Augmented GNNs since they incorporate LLMs to enrich graph inputs or guide GNN training, but retain a GNN backbone.

**LLM-BASED** methods include (a) GRAPHTEXT [53] that transforms $k$-hop neighborhoods into textual prompts and performs zero/few-shot classification using frozen LLMs and (b) RWNN [25] that converts random walks on graphs to node label anonymized sequendes and uses frozen LLMs for prediction. belong to this category due to their reliance on prompt-based inference using LLMs without any GNNs.

**GNN-BASED** methods include (a) ANYGRAPH [46] that pretrains a graph mixture-of-experts model using link prediction objective on diverse graphs that allows transfer to unseen datasets, (b) GRAPHANY [54] that learns permutation-invariant attention over a bank of pretrained LinearGNNs; (c) MDGPT [49] pretrains a GCN on multiple datasets with SVD-projected features and prompt vectors; (d) GCOPE [52] constructs a universal pretraining graph with virtual nodes and uses contrastive learning to train a shared GNN; (e) GPPT [38] introduces task-specific graph prompts for node task and link-prediction alignment; (f) GPROMPT [19] utilizes prompt vectors into graph pooling via element-wise multiplication (g) ALL-IN-ONE [39] combines token graphs with original graph as prompts (h) GPF [11] introduces prompt tokens and GPF-PLUS trains multiple independent basis vectors and combines them using attention (i) ULTRA [16] learns transferable graph representations by conditioning on relational interactions. (j) SCORE [44] introduces zero-shot reasoning on knowledge graphs using graph topology. All of these are grouped under GNN-BASED baselines as they rely on pretraining GNNs (often with auxiliary components like prompts or experts) to enable generalization to new graphs.

### D.2 Using SVM on the Pre-trained Representations

To assess the linear separability and structural quality of the learned graph representations from ALL-IN, we evaluate downstream graph classification accuracy using support vector machines (SVMs) with both linear and radial basis function (RBF) kernels (Table 3). This setup allows us to probe how well the learned representations support simple (linear) versus more expressive (nonlinear) decision boundaries.

We compare against several non-learnable baselines that do not involve any representation learning:

(a) Input Features ($\boldsymbol{X}$): Raw input features of each graph, computed by averaging node features.

(b) Propagated Input Features ($\boldsymbol{AX}$): Features after one round of neighborhood propagation, capturing local graph structure.

(c) Input Features along with random walk structural encodings ($\boldsymbol{X} \oplus \boldsymbol{S}$): Concatenates the raw features with random walk structural encoding (RWSE) [9], which encodes graph structure based on transition probabilities of random walks.

These baselines serve as direct input replacements for ALL-IN and are shared across both kernel settings. They provide a strong reference for understanding the inherent structure in the input space, independent of any learning or pretraining.

For ALL-IN, we report results both with and without concatenation of the input features to assess the added value of structural information in the learned embeddings.

Table 3: Graph classification accuracy (%) using SVMs with Linear and RBF kernels. Baselines are shared across both kernels. Results are reported as mean $\pm$ standard deviation over 10 runs.

| Method | MUTAG (ACC ↑) | PTC (ACC ↑) | PROTEINS (ACC ↑) | NCI1 (ACC ↑) | NCI109 (ACC ↑) | ENZYMES (ACC ↑) |
|---|---|---|---|---|---|---|
| **LINEAR SVM** | | | | | | |
| Input Features | 81.87 ± 7.25 | 60.88 ± 1.83 | **72.68 ± 0.58** | 64.59 ± 1.24 | 63.36 ± 2.22 | 22.00 ± 4.46 |
| Propagated Input Features | 69.64 ±14.21 | 57.34 ±10.89 | 59.56 ± 3.94 | 64.16 ± 1.22 | 63.26 ± 1.63 | 14.33 ± 5.01 |
| Input Features + RWSE | 80.96 ± 0.89 | 60.14 ± 1.15 | 65.74 ± 0.43 | 64.30 ± 0.16 | 63.45 ± 0.20 | 27.00 ± 4.63 |
| ALL-IN | 74.47 ± 7.70 | 53.12 ± 9.09 | 60.91 ± 4.25 | 63.26 ± 1.36 | 63.19 ± 1.89 | 21.16 ± 6.28 |
| ALL-IN + Input Features | 74.47 ± 7.70 | 52.84 ± 9.03 | 62.00 ± 4.29 | 64.45 ± 1.48 | 63.72 ± 1.67 | 21.50 ± 5.18 |
| **RBF SVM** | | | | | | |
| Input Features | 72.73 ±14.29 | 55.88 ±11.58 | 71.06 ± 2.93 | 66.44 ± 1.43 | 66.80 ± 1.35 | 33.33 ± 4.77 |
| Propagated Input Features | 79.70 ±11.03 | 54.10 ±10.25 | 72.05 ± 4.70 | 55.66 ± 5.80 | 58.05 ± 5.42 | 33.16 ± 4.43 |
| Input Features + RWSE | 79.21 ±10.99 | 58.71 ± 8.76 | 67.21 ± 6.22 | 70.68 ± 2.60 | 67.82 ± 2.79 | 36.66 ± 5.96 |
| ALL-IN | 82.98 ± 7.76 | 59.28 ± 9.13 | 70.62 ± 4.53 | 65.88 ± 1.62 | 65.68 ± 1.90 | 28.83 ± 5.87 |
| ALL-IN + Input Features | **84.06 ± 6.61** | **59.88 ± 7.72** | 71.42 ± 4.29 | **67.54 ± 1.33** | **67.34 ± 1.51** | **32.16 ± 6.71** |

Under the RBF kernel, ALL-IN combined with input features achieves the best performance on four out of six datasets, including PTC, NCI1, NCI109, and ENZYMES, highlighting its ability to encode discriminative patterns suitable for nonlinear classification. In contrast, performance under the linear kernel is more mixed, with RWSE showing strong results on datasets like PROTEINS, indicating some inherent linear separability in the structural baseline. Overall, these results demonstrate that ALL-IN learns representations that are expressive and transferable across diverse graph datasets, especially when paired with nonlinear classifiers.

### D.3 Additional results on transferability to unseen datasets

In Table 4, we present comparison with more baselines on our graph classification datasets MUTAG and PROTEINS. We describe below the changes we make to the following baselines to make them applicable to this setting:

- **GLEM-LM** [8]: This is a method that only supports tasks on text-attributed graphs. Since the TU Datasets [32] do not have node text attributes, we describe the input node features and pass them to ChatGPT.

- **GCOPE** [52]: This method introduces one virtual node for each node classification dataset, connecting it to all the nodes within the dataset. To perform graph classification, we introduce one virtual node for each graph classification dataset and connect it to all the nodes in all the graphs within the dataset.

- **ANYGRAPH** [46]: This method performs node classification by adding one node per class and connecting each training node to its corresponding class node. Classification of unlabeled nodes is performed by computing the dot product between the node's embedding and each class node embedding to rank the classes. To extend this paradigm to graph classification, we introduce a virtual node that connects to all nodes in the graph and add one class node per category. For classifying new graphs, we compute the dot product between the virtual node embedding and each class node embedding to rank the classes.

We leave out the following methods and provide justification below:

- **GRAPHTEXT** [53]: While the authors mention that GRAPHTEXT is applicable for graph classification, they do not provide a way to construct a graph syntax tree for an entire graph, which can be ambiguous as it could involve introducing a virtual node or averaging results from syntax trees of multiple nodes.

- **GRAPHANY** [54]: This method is explicitly only designed for node classification on arbitrary graphs as it relies on an analytical solution that is not directly applicable to graph-level tasks.

The results in Table 4 further substantiate ALL-IN's strong performance. These findings reinforce the observations made in the main paper: ALL-IN, with its frozen pre-trained encoder and a retrained head, effectively generalizes to new graph classification datasets with novel input features, surpassing a wide variety of adapted baselines.

Table 4: Performance on unseen graph-classification datasets with new input features. ALL-IN demonstrates strong transferability, underscoring its versatility and ability to handle different feature spaces. [†] indicates these methods were modified to work on these datasets, as explained in Appendix D.3

| Dataset | MUTAG (ACC ↑) | PROTEINS (ACC ↑) |
|---|---|---|
| **SUPERVISED BASELINES** | | |
| MLP | 67.20 ± 1.00 | 59.20 ± 1.00 |
| GIN [47] | 89.40 ± 5.60 | 76.20 ± 2.80 |
| **LLM-AUGMENTED GNNs** | | |
| OFA [31] | 61.04 ± 4.71 | 61.40 ± 2.99 |
| GLEM-LM[†] [8] | 72.97 ± 0.00 | 43.22 ±12.01 |
| **LLM-BASED** | | |
| RWNN-DeBERTa [25] | 58.22 ± 0.24 | 67.85 ± 0.53 |
| **GNN-BASED** | | |
| GCOPE[†] [52] | 81.87 ± 7.26 | 71.84 ± 3.48 |
| ANYGRAPH[†] [46] | 75.61 ± 6.94 | 72.23 ± 4.63 |
| MDGPT [49] | 57.36 ±14.26 | 54.35 ±10.26 |
| GPPT [38] | 60.40 ±15.43 | 60.92 ±12.47 |
| ALL-IN-ONE [39] | 79.87 ± 5.34 | 66.49 ± 6.26 |
| GPROMPT [19] | 73.60 ± 4.76 | 59.17 ±11.26 |
| GPF [11] | 68.40 ± 5.09 | 63.91 ± 3.26 |
| GPF-PLUS [11] | 65.20 ± 6.94 | 62.92 ± 2.78 |
| ULTRA(3G) [16] | 63.33 ± 0.00 | 58.09 ± 0.00 |
| SCORE [44] | 85.33 ± 2.11 | 68.54 ± 1.47 |
| ALL-IN (0 props) | 92.50 ± 6.60 | 76.72 ± 3.19 |
| ALL-IN | 92.90 ± 6.34 | 78.20 ± 3.81 |

## D.4 Asymptotic Computational Complexity

For a graph with $n$ nodes and $m$ edges, with node feature matrix $\boldsymbol{X} \in \mathbb{R}^{n \times d}$, projecting features using a random linear transformation takes $\mathcal{O}(ndh)$ time and $\mathcal{O}(nh)$ memory, where $h$ is the projection dimension. Computing $\{\boldsymbol{R}^{(p)}\}_{p=1}^{k}$ takes $\mathcal{O}(k(m+n))$ time, as this is equivalent to $k$ message-passing layers propagating $\boldsymbol{R}^{(0)}$. The centering operation takes $\mathcal{O}(knh)$ time.

When explicitly constructing the node-covariance operators $\boldsymbol{K}^{(p)} = \frac{1}{h}\boldsymbol{R}_c^{(p)}(\boldsymbol{R}_c^{(p)})^\top \in \mathbb{R}^{n \times n}$, the computational complexity is $\mathcal{O}(kn^2 h)$ and memory complexity is $\mathcal{O}(kn^2)$ (as $p = 1, \cdots, k$), resulting in quadratic complexity with respect to the number of nodes. This explicit construction is necessary in certain scenarios such as subgraph GNNs where the full pairwise similarity matrix is required as the graph structure itself [1, 14]. However, for standard message passing operations in most MPNNs [26, 47, 35], we can avoid explicitly constructing the covariance matrix. Since message passing can be written as a left-hand multiplication by a propagation matrix (our covariance operator $\boldsymbol{K}$), and by substituting the definition $\boldsymbol{K} = \boldsymbol{R}\boldsymbol{R}^\top$, we can compute $\boldsymbol{R}(\boldsymbol{R}^\top \boldsymbol{H}^{(\ell-1)})$ instead of $(\boldsymbol{R}\boldsymbol{R}^\top)\boldsymbol{H}^{(\ell-1)}$. This way, at no point do we need to hold the full covariance matrix in memory. This approach has computational complexity $\mathcal{O}(k(m + nh \cdot h^{(\ell-1)}))$ and memory complexity $\mathcal{O}(n(h + h^{(\ell-1)}))$ for the entire layer computation, where $h^{(\ell-1)}$ is the feature dimension of $\boldsymbol{H}^{(\ell-1)}$, avoiding the $\mathcal{O}(n^2)$ memory bottleneck while producing mathematically identical results.

Therefore, the computational complexity of ALL-IN depends on the specific use case, i.e. it is $\mathcal{O}(k(n^2 + m))$ time and $\mathcal{O}(kn^2)$ memory when explicit covariance matrices are required and $\mathcal{O}(k(m + nh \cdot h^{(\ell-1)}))$ time and $\mathcal{O}(n(h + h^{(\ell-1)}))$ memory for standard MPNN message passing.

## E   Dataset Information

In this section, we describe the datasets used in our experiments. We categorize them based on their use in pretraining and task transferability.

Table 5: Statistics of pre-training datasets used in ALL-IN. The datasets span molecules, drugs, computer vision-derived graphs and 3D shape point clouds.

| Dataset | # Nodes | # Edges | # Features | # Classes | Domain / Category |
|---|---|---|---|---|---|
| ZINC | 23.2 (avg) | 24.9 (avg) | 28 | - | Molecular Graph Regression |
| OGBG-MOLHIV | 25.5 (avg) | 27.5 (avg) | 9 | 2 | Drug Discovery |
| OGBG-MOLESOL | 13.3 (avg) | 13.6 (avg) | 9 | - | Solubility Prediction |
| OGBG-MOLTOX21 | 18.6 (avg) | 19.4 (avg) | 9 | 12 (multi-label) | Toxicology |
| MNIST (SUPERPIXELS) | 75 | 142 | 1 | 10 | Vision (Digits) |
| CIFAR10 (SUPERPIXELS) | 85 | 170 | 1 | 10 | Vision (Objects) |
| MODELNET | 100 (fixed) | 150 (fixed) | 3 | 40 | 3D Shape Classification |
| CUNEIFORM | 62 (avg) | 150 (avg) | 1 | 30 | Archaeology / OCR |
| MSRC 21 | 212 (avg) | 336 (avg) | 4 | 21 | Image Segmentation |

Table 6: Statistics of finetuning datasets used in our experiments. For node classification datasets (citation networks), we report the total number of nodes and edges. For graph classification datasets (bioinformatics), we report the number of graphs and average graph sizes.

| Dataset | # Graphs / Nodes | # Edges | # Features | # Classes | Domain / Task |
|---|---|---|---|---|---|
| CORA | 2,708 nodes | 5,429 | 1,433 | 7 | Citation Network / Node Classification |
| CITESEER | 3,327 nodes | 4,732 | 3,703 | 6 | Citation Network / Node Classification |
| PUBMED | 19,717 nodes | 44,338 | 500 | 3 | Citation Network / Node Classification |
| MUTAG | 188 graphs | 17.9 (avg) | 7 | 2 | Bioinformatics / Graph Classification |
| PROTEINS | 1,113 graphs | 39.1 (avg) | 3 | 2 | Bioinformatics / Graph Classification |

## E.1 Pre-training Source Datasets (A1)

For pretraining ALL-IN, we use 10 diverse datasets covering molecular graphs, drugs, computer vision, and 3D shapes. The statistics for each dataset are summarized in Table 5. The detailed information is as follows:

- **ZINC** [10] is a molecular property prediction dataset where the task is regressing the constrained solubility values of molecules. We report mean absolute error (MAE) as the evaluation metric.

- **MOLHIV, MOLESOL, MOLTOX21** [22] is a collection of molecular graphs from the OGB benchmark covering drug discovery and toxicity prediction tasks. Depending on the dataset, we perform binary classification (MOLHIV), regression (MOLESOL), or multi-label classification (MOLTOX21). Performance is measured using ROC-AUC or RMSE, as appropriate.

- **MNIST, CIFAR10** [10] are computer vision datasets converted into graph-structured superpixels. Each image is modeled as a fixed-structure graph, with 1-dimensional input features and a 10-way classification objective.

- **MODELNET** [45] is a 3D object classification benchmark where shapes are represented as fixed-size point cloud graphs. We use the 10-class subset.

- **CUNEIFORM** [32] is a graph-based OCR dataset derived from ancient script symbols, consisting of 62-node graphs with 150 edges on average and a 30-class prediction target.

- **MSRC-21** [32] is an image segmentation dataset where region adjacency graphs are constructed from visual scenes. Each graph has approximately 212 nodes and 336 edges, with 4-dimensional node features and 21 semantic class labels.

## E.2 Transferability to Unseen Datasets and Input Features (A2)

To evaluate the transferability of ALL-IN to unseen input features, we choose the following datasets summarized in Table 6 and explained below:

- **CORA, CITESEER, PUBMED** [48]: In these datasets, nodes represent academic papers and edges denote citation links. Each node is assigned a class label corresponding to a subject area. The task is to predict the category of a paper based on its content features and citation graph. Models are evaluated under transductive learning settings using fixed splits [48].

Table 7: Hyperparameter Configuration for Pretraining Stage.

| Category | Hyperparameter (Value) |
|---|---|
| **Architecture** | |
| Activation Function | ReLU |
| Attention Type in GPS | PerformerAttention |
| GPS Heads | 4 |
| Channels $h^{(\ell)}$ | 256 |
| Random Projection Dim $h$ | 512 |
| Backbone GNNLayer | gps_gine |
| Number of Layers $L$ | 6 |
| Input PE Dim $h_s$ | 20 |
| Use Random Projections | True |
| # Node-Covariance Operators $k$ | 0, 2 |
| **Training Setup** | |
| Pretraining Epochs | 500 |
| Batch Size | 64 |
| Dropout | 0.0 |
| Learning Rate | 0.0001 |
| Weight Decay | 0.0 |
| Normalization Type | batchnorm |

- **MUTAG** [32]: A binary classification dataset of small molecule graphs. Nodes represent atoms with categorical features, and graphs are labeled based on mutagenic effect on a bacterium.
- **PROTEINS** [32]: A dataset of protein structures modeled as graphs where nodes represent secondary structure elements and edges reflect neighborhood in the amino acid sequence. Each graph is labeled as enzyme or non-enzyme.

# F  Implementation Details

We implement ALL-IN using PyTorch [33] (BSD-3 Clause license) and PyTorch Geometric [13] (MIT license). For experiment tracking and hyperparameter logging, we utilize the Weights and Biases framework [2]. Experiments were conducted with NVIDIA RTX A6000, RTX 4090, and NVIDIA A100 GPUs.

For all experiments, we use the GPS framework [35] with the GIN message passing layer [47] for $\{\text{GNNLayer}^{(\ell,\boldsymbol{A})(\cdot,\boldsymbol{A})}\}_{\ell=0}^{L}$, and we use standard message passing layer for other operators.

## F.1  Pre-training on Different Source Datasets (Q1)

To evaluate large-scale transfer, we pretrain ALL-IN on a diverse set of 10 graph datasets spanning multiple domains, as described in Appendix E. Each training epoch cycles through all datasets once, optimizing dataset-specific objectives. We train for 500 epochs and checkpoint every 25 epochs. Hyperparameters are detailed in Table 7. To accelerate training, (1) we use `DataParallel` to support multi-GPU runs, (2) cache the random projection matrix $\boldsymbol{C}$ and refresh every 100 steps, (3) sample 10,000 graphs randomly at each epoch for MNIST and CIFAR10, and (4) sample 128 nodes with 6-nearest neighbors as edges for MODELNET in each graph.

## F.2  Evaluation on Unseen Datasets and Input Spaces (Q2)

To evaluate the transferability of ALL-IN to unseen datasets with novel input features, we freeze the pretrained encoder and evaluate its representations by training lightweight classifiers on new target datasets. These datasets span both node-level and graph-level classification tasks, with input feature spaces and labels disjoint from those used during pretraining.

For each target dataset, we instantiate a prediction head using one of the following: (1) a **multi-layer perceptron (MLP)** for both node and graph classification tasks; (2) a **2-layer GCN** [26] applied to node classification benchmarks (CORA, CITESEER, PUBMED); and (3) a **2-layer GIN** [47] for graph classification benchmarks (MUTAG, PROTEINS). All prediction heads are trained with frozen ALL-IN features as input. No gradients are backpropagated into the encoder during this stage.

For MLPs, we use a single hidden layer of size 128 with ReLU activation, followed by a softmax or sigmoid output layer, depending on whether the task is single-label or multi-label. We train all classifiers using the Adam optimizer with a learning rate of 0.001 and early stopping based on validation loss. Node classification models are trained on the standard 20/30/50 splits [48] and evaluated using accuracy. For graph classification, we perform 10-fold stratified cross-validation and report the mean and standard deviation of classification accuracy.

All transfer experiments are implemented in PyTorch and PyTorch Geometric. Environment and optimization settings match those described in Appendix F.1.

# G   Broader Impact

This work proposes a method for learning transferable representations for graphs with different input feature spaces. It may benefit tasks where training data is limited or diverse in structure, such as molecule classification or drug discovery. As the method is task-agnostic, it could be applied in various domains without careful tuning. We do not foresee any immediate negative societal impact, but users should be cautious when applying the method to sensitive domains without domain-specific validation.

