# OpenReview forum: "All In: Bridging Input Feature Spaces Towards Graph Foundation Models"
_NeurIPS.cc/2025/Workshop/UniReps — UniReps2025_

### Official Review · Reviewer_fxS1 · 2025-09-05
**Thank you for the interesting paper.**

**Confidence:** 5

**Review:**

**[Summary]**

This paper proposes a technique the authors call ALL‑IN, which is a framework to decouple graph representation learning from the original node feature space to form a unified second order representation across graphs. The technique roughly proceeds by (i) randomly projecting node features to a shared h‑dimensional space; (ii) constructing node–node covariance operators from propagated projections; (iii) feeding parallel GNN branches with the aforementioned operators; (iv) concatenating their outputs. This makes the method distributionally invariant to feature permutations as well as making its expected operator invariant to orthogonal transforms. Additionally, asymptotic consistency is attained as $h \to \infty$, which the authors explicitly show in the appendix. Empirically, one encoder pre‑trained across datasets seems to transfer to unseen datasets when training only a small prediction head, which appears to achieve competitive results compared to SOTA in some cases.

**[Strengths]**

- I love the idea presented in Section 2 (e.g., lines 36-39), where IID isotropic gaussian projections are employed to enforce distributional invariance. This is clever.

- An isotropic gaussian matrix is known to be approximately orthogonal (in the full-rank case). Intuitively, the projection tends to do an orthogonal rotation to the feature vectors (with some noise) at each term, which ensures that the network learns to denoise from the covariance features. This would help with robustness.

- The theory is valid and neat. Conceptually, I find the idea simple and easy to understand, even considering some scalability issues.

- Competitive transfer from pre-training seems to be a strong benefit of this technique. Congratulations.

**[Weaknesses]**

- Explicitly using covariance makes this a second order method. Typically, that tends to cause many scalability difficulties in the framework, even with projections into a lower dimensional space as is done in the paper. I should note that the authors are explicit about this shortcoming, as it is stated in lines 129-132.

- Despite the "competitive with SoTA" framing taken in 3.1, Table 1 shows that there are tasks in which ALL-IN performs significantly worse than existing methods. However, I still believe this work is significant, as being a general all-purpose (and not task-specific) method, the first step would be to demonstrate non-degradation of capabilities contrasted with per-task models. That it performs much worse on some tasks is not surprising in my view.

**[Originality]**

Good. There are some related works, but they are well cited in Appendix A. Similar projection-based GFM methods were presented at ICML 2025, but none of them explicitly formed the projected covariance matrix to my knowledge.

**[Clarity]**

Yes, the paper is clear and easy to read.

**[Soundness]**

The theory and experiments seem sound. I have not reviewed their code, however.

**[Quality of Results]**

I rate this as good. Generally speaking, I agree that the experiments show ALL-IN is competitive with some baselines.

**[Significance]**

Moderate to good. As noted in the limitations, if the scalability issue due to second-order tensor formation (i.e., covariance compute per layer) could be resolved, this would be a strong and much needed contribution. At present, higher-order operator $K^{(p)}$ formation is very expensive.

**[General Review]**

Not much to say here. I enjoyed reading the paper.

**[Questions/Suggestions]**

S1) It might be helpful to bold-face the best performing entries in Table 1. At the moment, it doesn't jump out. It appears there are some benchmarks where ALL-IN performs significantly better than other baselines, but some where ALL-IN performs much worse.

Q2) A major problem might be that unlike (some) adjacencies, the matrix operator sequence $K^{(p)}$ formed during deployment is almost surely dense due to the projection step. What techniques might the authors suggest to enhance scalability?

Q3) The main text claims that the gaussian projection operator $C$ is “sampled at each forward pass,” but Appendix F.1 states that $C$ is cached and refreshed every 100 steps. This is a significant and major difference. How sensitive are results to the refresh interval? I understand that due to the cost of forming such a projector, the authors went for caching, but even still, it would help to have this information.

S4/Q4) Have the authors considered Nystrom/sketching/landmark approximations or block‑sparsification of the second-order operators? This might help greatly with scalability of the proposed methods.

S5) It would be great to have some sort of explanation for the poor performance of ALL-IN on some datasets in Table 1.

S6) Theorem 2.1 (the theoretical contribution) would still hold without the gaussian projection step, as a closer look at the proof confirms. The theorem generalizes the idea that orthogonal transformations do not affect the covariance, which is a very well-known fact. The added value here is coupling this with (a) isotropic random projections to obtain distributional invariance to permutations and (b) stochastic training that aids symmetry‑breaking and regularization. The latter two are new, and in my view, generally hidden away in the Appendix (especially for (b)), and I think it could help to have this emphasized more in the main paper. The authors do mention (a) in the main text, but it might be helpful to emphasize this a little more. Why is (b) hidden in the Appendix–was it meant to be a latent treat for reviewers who actually read your proofs? I think this is a nice result.

**[Typos/Minor]**

- line 10, "retraining , pointing" (extra space)

- line 35-36, equation (1) states "sampled at each forward pass, (line break/line 36) sampled independently at each forward pass". This is rather repetitive.

- line 44, "where $S$ where $\ell$"

- Appendix B and C are very repetitive. Appendix C is essentially the same as Appendix B, only difference being that proofs have been added for the same stated theorems.

**[Final Recommendation]**

I recommend acceptance. This paper is clearly written, and provides a clean method for feature‑space‑agnostic graph learning with supportive theory and believable/reasonably strong cross‑dataset results. This approach feels particularly timely, as many papers in recent conferences have proposed graph foundation model architectures.

**Score:**

4

**Topic Fit:**

3

---

### Official Review · Reviewer_vumX · 2025-09-15
**Good intuitive ideas, and good results**

**Confidence:** 4

**Review:**

The paper proposes an approach to graph foundational models that can generalize to multiple unseen datasets without relying on retraining. It performs random projection on the features and builds a covariance-based node representation, and so, it is independent of feature ordering. Specifically, it builds higher-order covariance matrices that are used to propagate features

## Strengths
- The idea is intuitive, and the presentation is easy to understand
- Independence of node ordering and random projection allows for generalization to a large class of feature spaces
- The results are good, even if they do not exceed the state of the art.

## Weaknesses
- Theory: It would help if the authors could clearly state and explore the assumptions on the input. Since we are only working in the space of inner products and trying to be permutation invariant, perhaps the assumption is that the features are independent and are equally important?
- Algorithm: It was not clear how the random projection works during inference. In training, it is sampled for each forward pass. For inference, would you perform multiple samples and create an ensemble? How many samples are needed?
- Experiments: Running on a few more datasets and different splits will strengthen their claim and will further help explore the above point regarding the feature space. Understandably, this is a heavy task, but something to consider for a future longer version.

**Score:**

4

**Topic Fit:**

3

---

### Official Review · Reviewer_ChMp · 2025-09-17
**Addresses the challenges of heterogeneous input feature spaces in graph learning**

**Confidence:** 3

**Review:**

This paper tackles a hard problem in graph learning, how to deal with very different input feature spaces. Proposes a simple but effective solution using random projections and covariance operators. I like that it shows good results on many different datasets and even works on new ones without retraining, which makes it feel broadly useful. The theory gives it a solid base, and the experiments back it up well. The main downsides are that it hasn’t been tested on very large graphs yet, but overall a good paper.

## Strengths ##
- Addresses a core bottleneck in building GFMs, heterogenous feature space
- Looks Novel
- Showcases strong generalization on unseen datasets without training the encoder
- Solid theoretical foundation
- Provides clean architecture

## Weakness ##
- Scalability is limited, not evaluated on large graphs
- covariance-based models might be prone to bias propagation
- Some datasets show minor improvements
- Requires dataset specific prediction heads, so not fully e2e universal

**Score:**

4

**Topic Fit:**

3